# Gas Hydrate Estimate in an Area of Deformation and High Heat Flow at the Chile Triple Junction

**Lucía Villar-Muñoz [1,\*], Iván Vargas-Cordero [2] , Joaquim P. Bento [3] , Umberta Tinivella [4] , Francisco Fernandoy [2,5] , Michela Giustiniani [4] , Jan H. Behrmann [1] and Sergio Calderón-Díaz [2]**

[1] GEOMAR Helmholtz Centre for Ocean Research, Wischhofstr. 1-3, 24148 Kiel, Germany; jbehrmann@geomar.de

[2] Facultad de Ingeniería, Universidad Andrés Bello, Quillota 980, Viña del Mar 2531015, Chile; ivan.vargas@unab.cl (I.V.-C.); francisco.fernandoy@unab.cl (F.F.); sergio.calderon@unab.cl (S.C.-D.)

[3] Escuela de Ciencias del Mar, Pontificia Universidad Católica de Valparaíso, Av. Altamirano 1480, Valparaíso 2340000, Chile; jnettojunior@gmail.com

[4] Istituto Nazionale di Oceanografia e di Geofisica Sperimentale (OGS), Borgo grotta gigante 42/c, 34010 Sgonico, Italy; utinivella@inogs.it (U.T.); mgiustiniani@inogs.it (M.G.)

[5] Centro de Investigación Para la Sustentabilidad (CIS), Universidad Andrés Bello, República 252, Santiago 8370134, Chile

[\*] Correspondence: lucia.villar@gmail.com; Tel.: +56-9-5226-4461

**Abstract:** Large amounts of gas hydrate are present in marine sediments offshore Taitao Peninsula, near the Chile Triple Junction. Here, marine sediments on the forearc contain carbon that is converted to methane in a regime of very high heat flow and intense rock deformation above the downgoing oceanic spreading ridge separating the Nazca and Antarctic plates. This regime enables vigorous fluid migration. Here, we present an analysis of the spatial distribution, concentration, estimate of gas-phases (gas hydrate and free gas) and geothermal gradients in the accretionary prism, and forearc sediments offshore Taitao (45.5°–47° S). Velocity analysis of Seismic Profile RC2901-751 indicates gas hydrate concentration values <10% of the total rock volume and extremely high geothermal gradients (<190 °C·km$^{-1}$). Gas hydrates are located in shallow sediments (90–280 m below the seafloor). The large amount of hydrate and free gas estimated (7.21 × 10$^{11}$ m$^3$ and 4.1 × 10$^{10}$ m$^3$; respectively), the high seismicity, the mechanically unstable nature of the sediments, and the anomalous conditions of the geothermal gradient set the stage for potentially massive releases of methane to the ocean, mainly through hydrate dissociation and/or migration directly to the seabed through faults. We conclude that the Chile Triple Junction is an important methane seepage area and should be the focus of novel geological, oceanographic, and ecological research.

**Keywords:** BSR; gas hydrate; methane; seepage; active margin; Chile Triple Junction

## 1. Introduction

Gas hydrate is a crystalline ice-like solid formed by a mixture of water and gasses, mainly methane, giving place to a clathrate structure [1,2] that can be stored in the pore space of marine sediments under low temperature (<25 °C) and high pressure (>0.6 MPa) conditions. Methane gas may be produced biogenically at shallow depths or may migrate from a deeper source through advective transport along pathways such as fracture networks, faults, or shear zones (e.g., [3]). Since the gas hydrates are rich in methane, 1 m$^3$ of hydrate will yield 0.8 m$^3$ of water and 164 m$^3$ of methane at standard pressure and temperature (STP: 0 °C, 0.101325 Mpa) conditions [4], and a significant amount of hydrate

represents unconventional and potential energy resources [5]. Moreover, gas hydrates play a part in global climate change, geo-hazards, and potential drilling hazards (e.g., [6–10]).

It is possible to identify gas hydrates in marine sediments using seismic profiles. The main indicator is the so-called Bottom Simulating Reflector (BSR), whose presence is related to the impedance contrast between high velocity gas hydrate-bearing and the underlying low velocity free gas layer [11–14]. Gas hydrate occurrences along the Chilean margin have been reported in many places by analysing the available seismic profiles (e.g., [11,14–27]), as well as more recently by direct identification of cold seeps emitting methane at the seafloor [28–34]. The first discovery of a seepage area was in 2004, offshore Concepción. Afterwards, other bathyal seep sites were identified, mainly by the presence of typical seep communities: (a) off the Limarí River at ~30° S (~1000 m water depth); (b) off El Quisco at 33° S (~340 m water depth); and c) off the Taitao Peninsula at ~46° S (~600 m water depth) [30–37].

Cold seeps sites are found in both active and passive margins and are related to the expulsion of methane-rich fluids. Chemosynthetic communities have been observed along active margins characterized by a well-developed accretionary prism, and along tectonically erosive margins [38]. The Chile Triple Junction (CTJ) area is a spectacular example of tectonic erosion (e.g., [39]). Even though many investigations are associated with seepage identification and gas expulsion quantification (gas bubbles) (e.g., [29,38,40–42]), there are few cases where the objective was to estimate the size of the gas source, as concentrations of gas hydrate and free gas [43].

Furthermore, the studies that report estimates of gas hydrates concentrations along the Chilean margin are scarce, even though, in the last decades, gas-phase concentrations have been estimated by fitting modelled velocity with theoretical velocity in the absence of gas [44]. These estimates reach an average of 15% and 1% of the total volume of gas hydrate and free gas concentrations, respectively [22,24,25,27]. A recent investigation of the southernmost Chilean continental margin showed that a regionally extensive methane hydrate reservoir, characterized by high gas hydrate and free gas concentrations, is present in the Patagonian marine sediments [27]. This could be an important natural resource for Chile, but because of the hydrate decomposition, this also potentially poses a great environmental threat.

On the other hand, the Chilean south-central margin is one of the tectonically most active regions on Earth, with very large and mega-scale earthquakes occurring every 130 and 300 years, respectively [45]. The margin segment close to the CTJ is characterized by high seismicity [46,47] that may trigger submarine sediments sliding and eventual gas hydrate dissociation. Some authors suggest that large subduction zone earthquakes have the potential to trigger hydrocarbon seepage to the ocean and possibly to the atmosphere (e.g., [29,48]). In this context, known gas hydrate quantities stored beneath marine sediments play an important role in the geohazard assessment. Besides, in subduction zones such as the Chilean margin, fluids play a key role in the nucleation and rupture propagation of earthquakes [49], and are a major agent of advective heat transfer from depth to the Earth's surface. For this reason, it is crucial to know the pathways where methane-rich fluids could migrate. The release of this methane stored in the forearc wedge could have consequences for the ocean and atmosphere systems, and the destabilized gas hydrate-bearing sediments are a formidable geohazard, in the form of submarine slumps, induced earthquakes, and tsunamis (e.g., [2,6,50–54]).

The particularity of the Chilean margin close to the CTJ, with anomalous heat flow and high seismicity, together with the presence of hydrothermal systems (e.g., [55]) and possible seafloor seeps, offers a unique scenario to study hydrate deposits. The aim of this study is to characterize and estimate the methane concentrations (hydrate and free gas phases) stored in the marine sediments in order to understand the potential amount of this gas that could be released through these natural pathways, likely affecting the geochemical properties of the seawater and, consequently, the marine ecology.

*Geological Setting*

The CTJ (Figure 1) is the site of the intersection of three tectonic plates: Nazca, Antarctic, and South America [39,56,57]. Here, the Chile Rise (CR), an active spreading centre, is being subducted beneath the South American continental margin. Ridge subduction began near Tierra del Fuego ~14 million years ago (Ma) and then migrated northwards to its current position north of the Taitao Peninsula (e.g., [15]). The Nazca plate subducts beneath South America in an ENE direction at a rate of about 70 km·Ma$^{-1}$ north of the CTJ, and the Antarctic plate subducts in an ESE direction at about 20 km·Ma$^{-1}$ south of the CTJ (e.g., [56]). The CR spreading rate has been estimated to have been about 70 km·Ma$^{-1}$ over the past 5 Ma, but within the last 1 Ma, it has slowed down to about 60 km·Ma$^{-1}$ (e.g., [58]).

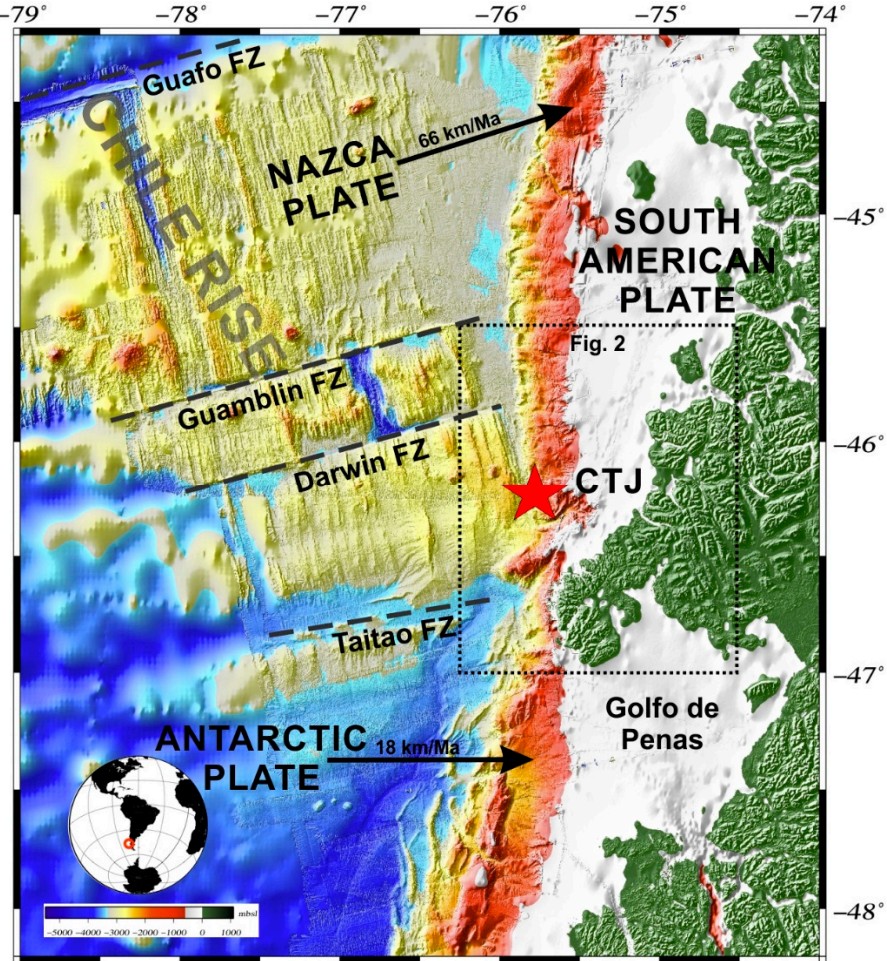

**Figure 1.** Location map of the study area offshore Taitao Peninsula. The bathymetry is based on GEBCO_08 Grid (version 20091120, http://www.gebco.net) and integrated with the IFREMER grid (cruise of the R/V L'Atalante, 1997). Tectonic setting of the Nazca, Antarctic, and South American plates: dashed black lines show the main Fracture Zones (FZ), red star marks a triple junction of the plates (CTJ), and dashed square corresponds to Figure 2.

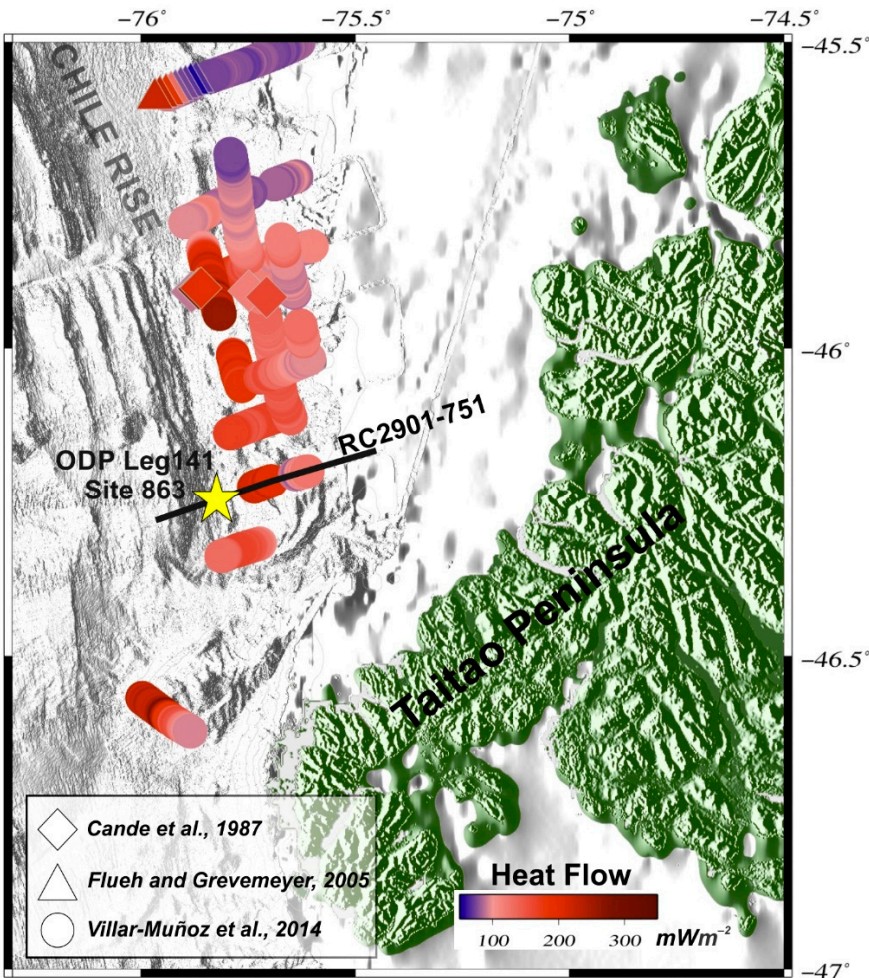

**Figure 2.** Heat flow (in m·Wm$^{-2}$) large-scale colour-coded based on BSR-derived heat flow and heat probes available for the area studied (after [26]). See text for description.

Close to the CTJ, the gas hydrate environment has peculiar characteristics relative to hydrate occurrences elsewhere. In fact, the ridge-trench collision perturbs pressure and temperature (PT) conditions within the sediment where hydrates have formed [11]. Excessively high heat flow, higher than 250 m·Wm$^{-2}$, was estimated above practically zero-age subducted crust (Figure 2). This is based on heat flow values derived from the depth of gas hydrate bottom-simulating reflectors [26,59] and direct measurements during the last decades [57,60].

The BSR-derived heat flow values are in general agreement with probe and borehole measurements [61]. Besides, high temperature gradients of 80–100 °C·km$^{-1}$ were obtained at the toe of the continental wedge (e.g., Site 863 in Figure 2), just above the subducted zero-age crust [55]. The thermal anomaly in the region varies rapidly due to the presence of a strong convective circulation [62].

More recently, explorative work at the seafloor close to the CTJ has provided evidence for a sediment-hosted hydrothermal source near (~50 km) a methane-rich cold-seep area [63]. Advective methane transport operates within 5 km of the toe of the accretionary prism [59,64]. However, in the interior regions of the wedge, free gas migration and in situ gas production (within the hydrate stability region) build-up the hydrate [15], and BSR-depth towards the trench appears to rise in the sediments in proximity of the spreading ridge [15,26].

Moreover, gas at the base of the hydrate layer at the CTJ could also be produced from hydrate dissociation when changes in PT conditions shift the zone of hydrate stability upward, not only due to the accumulation of overburden, but also due to changes in PT conditions associated with active ridge

subduction [11]. Increasing heat flow, associated with the approach of the CR, may have caused the base of the hydrate stability field to migrate ~300 m upwards in the sediments [15].

In this complex region, we find both active margin tectonic regimes: subduction erosion and subduction accretion occurring in close proximity (e.g., [65]). Bourgois et al [66] assumes that the tectonic evolution of the Chile margin in the area reflects the evolution of the tectonic regime at depth: subduction erosion from 5–5.3 to 1.5–1.6 Ma, followed by subduction accretion since 1.5–1.6 Ma. [67], indicates that subduction accretion occurring today along the pre-subduction segment is linked to a dramatic post-glacial increase in trench sediment supply. From evidence found by drilling at Ocean Drilling Program (ODP) Site 863 (Figure 2) at the CTJ proper, it was concluded that accretion ceased in late Pliocene, and presently, the small frontal accretionary prism is undergoing tectonic erosion [39,55].

## 2. Materials and Methods

### 2.1. Database

The analyzed seismic line was acquired in 1988 onboard the vessel R/V Robert Conrad within the framework of the project entitled "Paleogene geomagnetic polarity timescale" for Empresa Nacional del Petroleo (ENAP). The seismic profile was acquired using an air gun array with a size of 0.062 m$^3$. The shot spacing was approximately 50 m, and the streamer length was 3000 m and included 236 channels with an intertrace of 12.5 m. The seismic line RC2901-751 analyzed in this study was modelled to estimate gas hydrate and free gas concentrations.

During ODP Leg 141, the Site 863 located a few km south of the CTJ was drilled along the profile RC2901-751 in an area where the axis of the spreading ridge is subducting at 50 ka (Figure 2). Porosity and temperature data were obtained from this site.

### 2.2. Methods

The processing was performed using open source Seismic Unix software and codes ad-hoc [68] and includes a tested method reported in several studies [14,22,24,25,27,43,69]: (a) BSR identification, (b) seismic velocity modelling, (c) gas-phases estimates, and (d) geothermal gradient estimation.

(a) BSR identification: a stacking section was obtained by using standard processing (i.e. geometry arrangement, spherical divergence, velocity analysis, normal-moveout corrections, stacking, and filtering). The objective was to identify the BSR in a selected part of the stacking section. Once the BSR was recognized, the seismic velocity was modelled.

(b) Seismic velocity modelling: An in-depth velocity model was obtained using the Kirchhoff Pre-stack Depth migration (PreSDM) iteratively with a layer stripping approach (details in [70,71]). This approach uses the output of the PreSDM, the common image gathers (CIGs) [71]. In the seismic profile, three layers were modelled: the first between the seawater level and the seafloor reflector (SF layer); the second between the seafloor and the BSR (BSR layer); and the third between the BSR and the Base of Free Gas (BGR layer). It started with an initial constant velocity model equal to 1480 ms$^{-1}$. After four iterations, the SF reflector in the CIGs was flat, suggesting that the migration velocity was correct. The correct migration for BSR and BGR was reached after 25 and 15 iterations, respectively. Below the BGR, a velocity gradient was included and, to improve the migration result, the final velocity model was smoothed. Finally, band-pass filtering and mixing were applied to improve the final PreSDM image. The sensitivity was considered a depth error equal to 2.5% proposed by [22] after a sensitive test.

(c) Gas-phases estimates: Once the final velocity model had been built, it was converted into gas hydrate and free gas concentrations. At first, a qualitative estimate was performed, comparing the modelled velocity curves against theoretical curves in the absence of gas. Afterwards, positive anomalies were associated with gas hydrate presence, while negative anomalies were related to free gas presence. Modified Hamilton's curves [72] were adopted to estimate the theoretical velocity curves in the absence of hydrates and free gas or full water saturated sediments [73]. Gas hydrates and free

gas concentrations were modified until the velocity model fitted the theoretical model, to obtain a quantitative estimate. The resultant is a concentration model in terms of total volume (for more details see [44]). Regarding the sensitivity, errors for gas hydrate and free gas estimates were assumed to be equal to 1.2% and 0.3% of volume, respectively. These errors were evaluated by [74], who performed a sensitive test to determine the influence of each parameter on the estimation of gas hydrate and free gas content. In fact, the main error was related to the assumptions of sediment properties.

(d) Geothermal gradient estimation: The geothermal gradient, indispensable to calculating the theoretical BSR-depth, was estimated using the following relation:

$$dT/dZ = (T_{BSR} - T_{SEA})/(Z_{BSR} - Z_{SEA}), \tag{1}$$

where BSR and seafloor depths ($Z_{BSR}$, $Z_{SEA}$) were extracted from the PreSDM section. Seafloor temperatures ($T_{SEA}$) were based on measurements from CTD data collected during ODP Leg 141 [75], while BSR temperatures ($T_{BSR}$) were based on the dissociation temperature-pressure function of gas hydrates [4]. Our estimation only considers methane because ethane concentration is negligible [22]. With regard to sensitivity, an error of depth equal to 2.5% was considered for seismic data [22].

## 3. Results

### 3.1. BSR Identification

The Kirchhoff PreSDM section (Figure 3) shows:

(a) A normal fault at a distance of 7 km representing the boundary between the lower and upper part of the continental rise and slope, respectively. Moreover, evidence of slip affecting the seafloor, as shallow faults and fractures, is registered from 8 to 15 km of distance;

(b) A strong and almost continuous BSR on the section that only gets weak or null where faults and fractures appear. Below the BSR, it is possible to recognize a weak but continuous reflector interpreted as BGR and, so, a free gas layer with a thickness of about 70 m;

(c) A variable depth of the BSR ranging between 80 and 150 m below seafloor (mbsf). The maximum depth of BSR was detected at about 2200 meters below sea level (mbsl) from 0 to 6 km, while the minimum depth (about 80 mbsf) was identified upwards (from 7 to 16 km). From 16 to 21 km of distance (in the "uplift part" of Figure 3), the BSR depth increases, reaching a depth of 150 mbsf.

### 3.2. Seismic Velocity Model

Above the BSR, a layer with a velocity ranging from 1650 to 1740 m/s was identified, while below the BSR, the velocity decreases from 1288 to 1550 m/s. Besides, below the BSR, the velocity decreases upwards (from 15 to 21 km of distance; see Figure 3 dark blue color), reaching its minimum value. An opposite velocity trend was observed above the BSR; in fact, when the velocity increases above the BSR (from 10 to 21 km of distance), the minimum velocity values are found below it. The BSR depth increases to the east, as shown by the velocity curves in Figure 3.

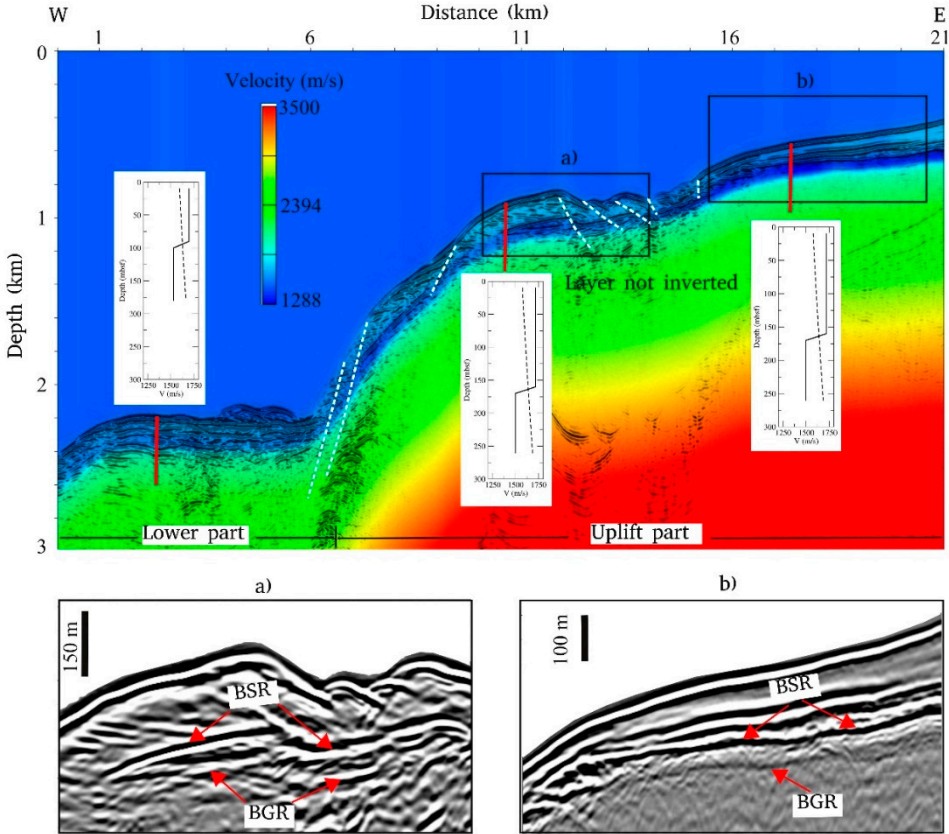

**Figure 3.** Velocity model superimposed in the Kirchhoff PreSDM section. The three inserts show the modelled velocity curves (solid black lines) and the theoretical curves in the absence of hydrates and free gas (dashed black lines) along the velocity model. Below, the rectangles indicate the position of the zooms in panel (a) and (b), in which red arrows indicate BSR and BGR (if present). The white dotted lines indicate faults and fractures.

### 3.3. Gas-Phases Estimates

High gas hydrates concentrations areas are located from 7 to 14 km of distance at approximately 1000 mbsl, reaching values ranging between 7 and 10% of total volume. Low gas hydrates concentrations regions (with values from 1 to 3% of total volume) are located from 1 to 6 km of distance at 2200 mbsl and from 15 to 20 km of distance at 600 mbsl (Figure 4). At shallow water depths, from 15 to 20 km of distance, high free gas concentrations were estimated, with values up to 0.8% of total volume. Note that hydrate and free gas concentrations show an opposite trend. In fact, from 7 to 14 km of distance, where gas hydrate concentrations increase (above the BSR), free gas concentrations decrease (see Top and Bottom panels in Figure 4). On the other hand, from 15 to 20 km of distance, where gas hydrate concentrations decrease, free gas concentrations increase.

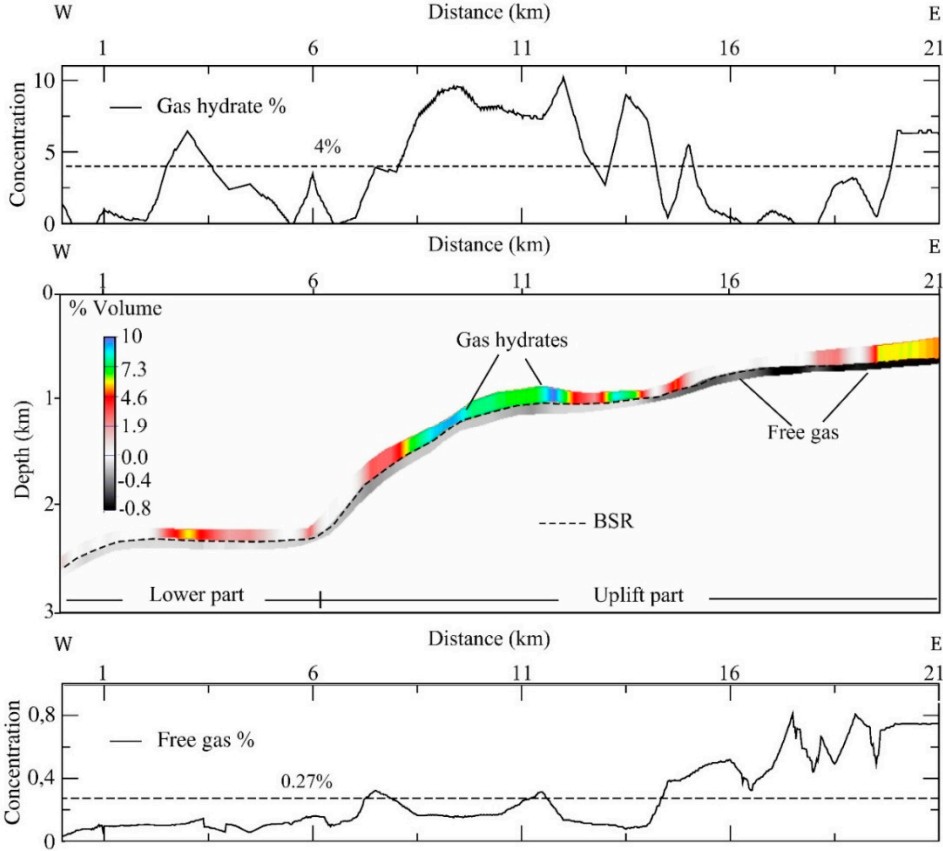

**Figure 4.** Gas hydrate and free gas concentration models and profiles relative to RC2901-751 seismic profile. Top panel: gas hydrate concentration values. Middle panel: gas-phase concentration model. Bottom panel: free gas concentration values. Dashed lines in the top and bottom panels correspond to the average gas hydrate and free gas concentrations, respectively.

### 3.4. Geothermal Gradient

The anomalous geothermal gradients calculated are variable in the seismic profile, ranging between 35 to 190 °C/km (Figure 5). The geothermal gradient increases towards the west (Figure 5), and the maximum values are at 2200 mbsl (see Figure 3). The minimum values were calculated on the east side of the profile (Figure 5) in correspondence of a water depth ranging from 600 to 1000 m. There are two isolated peaks (at ~9 and ~14 km of distance) of about 125 and 170 °C/km (Figure 5).

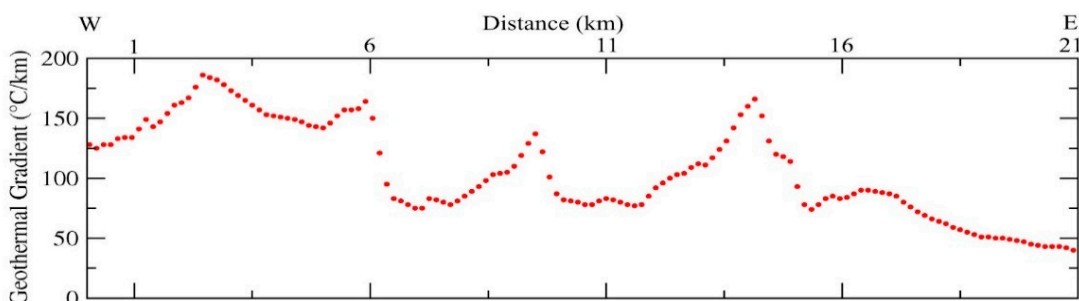

**Figure 5.** Geothermal gradient of the seismic profile RC2901-751. See text for details.

### 3.5. Gas Hydrate and Free Gas Volume at Standard Temperature and Pressure Conditions

In order to estimate the amount of methane stored in the marine sediments close to the CTJ region, bulk estimates of hydrate and free gas concentrations at standard temperature and pressure (STP) conditions were calculated using the following values:

- For gas hydrate: 4% of the total volume (dashed line in the upper panel of Figure 4), 50% porosity, thickness of the gas hydrate layer equal to 108 m, and a total projected area of about 2300 km$^2$. Considering these assumptions, the methane budget is $7.21 \times 10^{11}$ m$^3$ at STP conditions;

- For free gas: 0.27% of the total volume (dashed line in the lower section of Figure 4), 50% porosity, thickness of the free gas layer equal to 85 m, and a total projected area of about 2300 km$^2$. Considering these assumptions, the methane budget from gas hydrates is $4.1 \times 10^{10}$ m$^3$ at STP conditions.

The projected area was delimited based on the multi-resolution gridded Global Multi-Resolution Topography (GMRT) Synthesis [76] data and it comprises a part of the continental slope. The area was visually identified as the region that begins at the shelf break in the seaward edge of the shelf until it merges with the deep ocean floor at approximately 3000–3400 mbsl. All analyses were conducted with the open source Quantum Gis 3.4 (Qgis) and Generic Mapping Tools 5.4.4 (GMT) projects.

The free gas-volume expansion ratio was calculated using the Peng-Robinson equation of state [77], applying the methodology explained by [78]. Here, we assume that free gas is only composed of methane and it is located just below the gas hydrate stability zone. We divided the area containing free gas into five sub-areas to better assess the in-situ geothermal and pressure conditions and variations of the volume expansion ratios. Table 1 shows the free gas volume at in-situ and STP conditions. The rate of free gas volume expansion was calculated to estimate the volume of free gas content at STP conditions. The area was subdivided into five different regions, and at each one, pressure and temperature were calculated according to the corresponding geothermal gradient.

**Table 1.** Free gas volume at in-situ and STP conditions.

| Interval (mbsl) | Area (m$^2$) | Temperature (K) | Pressure (MPa) | Volume in-situ (m$^3$) | Volume STP (m$^3$) | Volume Expansion Ratio |
|---|---|---|---|---|---|---|
| 500–1000 | $4.19 \times 10^8$ | 285.8 | 7.6 | $4.25 \times 10^7$ | $3.69 \times 10^9$ | 86.8 |
| 1000–1500 | $4.37 \times 10^8$ | 289.7 | 12.7 | $4.44 \times 10^7$ | $6.71 \times 10^9$ | 151.2 |
| 1500–2000 | $5.59 \times 10^8$ | 291.0 | 17.7 | $5.68 \times 10^7$ | $1.20 \times 10^{10}$ | 212.2 |
| 2000–2500 | $3.69 \times 10^8$ | 291.5 | 22.8 | $3.75 \times 10^7$ | $9.90 \times 10^9$ | 263.9 |
| 2500–3000 | $2.79 \times 10^8$ | 292.1 | 27.9 | $2.84 \times 10^7$ | $8.67 \times 10^9$ | 305.5 |
| Total | $2.06 \times 10^9$ | | | $2.10 \times 10^8$ | $4.10 \times 10^{10}$ | |

## 4. Discussion

The seismic section showed evidence of active tectonics; in fact, a large normal fault zone located at 7 km of distance represents the boundary between the western and eastern sectors (Lower and Uplift part in the Figure 3). Morphological features close to the normal fault can be associated with active tectonic extension and uplift processes above the subducting CR seafloor spreading centre [39]. Further upslope deformation is characterized by normal faults and fractures with small offsets affecting shallow sediments (Figure 3). The weak seismic character of BSR in the seaward (westward sector) is related to low free gas concentrations, while in the uplifted landward (eastward sector), a continuous and strong BSR can be related to high free gas concentrations up to 0.8% (Figure 4). These values are consistent with free gas concentrations reported by [11] along Seismic Line 745, located northward of this study area. A shallow BSR depth (average ~100 mbsf) can be explained by a high heat flow (average > 200 mW/m$^2$) and geothermal gradient (average ~90 °C/km), as reported by [26] and in agreement with this study. In addition, vertical and lateral velocity variations above and below the BSR can be associated with gas hydrate and free gas presence and their concentration changes. Maximum velocity values above the BSR (up to 1740 m/s) can correspond to high gas hydrate concentrations, whereas low velocities below the BSR (around 1290 m/s) are related to high free gas concentrations (Figures 3 and 4). In fact, this low velocity can only be explained with free gas presence.

The gas-phase concentration distribution is in general agreement with heat flow reported by [26]. Moreover, low concentrations of gas hydrate and free gas coincide with high values of heat flow and geothermal gradients close to the Chile trench and the plate boundary (Figures 2 and 5), while

high concentrations of gas hydrate and free gas are associated with a low heat flow and geothermal gradient further up the continental slope. A similar pattern was also recognized by [22] on the Chilean continental slope around 44° S.

The observation that both gas hydrate and free gas concentrations in the sediments have lower values close to the trench in the CTJ area could be explained as a result of gas hydrate dissociation and free gas migration in a regime of fluid advection under high heat flow conditions [59]. High heat flow is caused by the subduction of the Chile Rise [11,26,57,59,60], and geothermal fluids are supplied from deeper strata [67] that are undergoing deformation, anomalous compaction, and de-watering (e.g., [55,65]). The highest values of heat flow are located close to the heat source near the trench (Figures 2 and 5). We assume that in this area, the advective heat transfer in a regime of rising heat flow can change the pressure-temperature conditions, causing gas hydrate dissociation in the past and likely in the present. Low concentrations of free gas close to the trench (~0.1% of total volume), can be explained due to a variable production. Here, the dissociated hydrates is released as free gas and can migrate up into the hydrate stability zone, giving place to gas hydrate formation in higher areas (from 7 to 14 km of distance in Figure 4), increasing gas hydrate concentrations (~8% of total volume). However, active faults and fractures in the lower forearc can destroy stratigraphic seals and, consequently, impede free gas storage (e.g., [26,43]) above the subducting spreading ridge. This may explain the low concentrations of gas hydrate and free gas layers calculated close to the trench and high concentrations in shallow waters, where the lower values of heat flow were found, and deformation is less prevalent (Figures 3 and 4). Note, however, that low concentrations of gas hydrate and free gas were also found close to faults and fractures because of the enhancement of fluid-escape (Figures 3 and 4). Therefore, high heat flow due to spreading ridge subduction, tectonic faulting, and vigorous fluid advection at the leading edge of the overriding South American plate may indeed be a major factor for hydrate and gas reservoir distribution offshore Taitao Peninsula. Moreover, the highest value of free gas concentration, located in the shallower part of the accretionary wedge (~16 km of distance; Figure 4), can be explained by the upward migration of gas towards an impermeable hydrate layer, forming a structural trap [22]. Note also that this sector is characterized by the absence of faults that could act as pathways for upward fluid migration.

The anomalous heat flow close to the CTJ changes the stable PT conditions for the gas hydrate, promotes its dissociation and fluid escapes. The dissolved methane from gas hydrates could enter into the ocean through fluid ventings or as gas bubbles [79]. Some of the dissolved methane is diluted and oxidized as it rises through the ocean interior. However, an increase in gas methane entering the ocean above seawater saturation could lead to methane reaching the ocean surface mixed layer and being transported to the atmosphere via sea-air exchange [80].

A question worth discussing here is whether some of the methane in gas hydrates in the lower continental slope may in fact have been formed by abiotic processes (e.g., [81]) during the formation of serpentinite from ultramafic rocks. This can be valid for hydrates present in sediments just above the youngest crust of the CR subducted (near the trench), where active serpentinization and methane venting can initiate, develop, and survive, as was observed in similar regions (e.g., [82]). ODP Site 863 (see Figure 2 and [55]) is located on the seismic line presented in this study, right above the subducting oceanic spreading ridge. Pore waters squeezed from the drill cores recovered at ODP Site 863 show very high pH values up to 10.5, especially at drillhole depths greater than 600 meters below the sea floor. Along with the concentration profiles of F, B, Cl, and $SO_4$, this suggests that the pore fluids could be created from a sequence of reactions involving Mg-depleted fluids (see Figure 6 and description on p. 406 of [55]). This can be taken as an indication of metasomatic alteration in the serpentinized peridotite of the oceanic mantle (e.g., [83,84]) belonging to the downgoing plate at depth. Recently, Suess et al. [85] has shown that gas hydrates involving abiotically formed methane might be formed in sediment drifts overlying altered oceanic crust and mantle in slow-spreading environments. It is possible to envisage a similar scenario here, with the difference that the sediments of the lowermost

continental slope are not directly sedimented above the spreading ridge, but are tectonically thrusted over the downgoing plate.

Finally, the estimated volume of gas hydrate calculated in the present study was lower than the values calculated in other regions along the Chilean margin (e.g., [27]). We hypothesize that this can be explained by the following reasons: (a) limited sediment accumulation due to the shortening of the wedge close to the CTJ, which causes unfavourable conditions for the formation of gas hydrates [11,39]; (b) the presence of faults and fractures that can locally promote fluid escape and prevent gas hydrate formation (e.g., [43,85]); (c) faults identified in the seismic profile (Figure 3) cross the transition layer of the gas hydrate phase and serve as pipes that drain water and methane to the seafloor (e.g., [85]); (d) the CTJ is characterised by an anomalous thermal state (e.g., [26]) that inhibits the formation of gas hydrates, by changing the gas hydrate stability zone.

## 5. Conclusions

The results of this research for the gas hydrate in the margin close to the Chile Triple Junction lead us to conclude that:

- The values for gas hydrate concentration are lower than 10% of the total rock volume. The highest concentrations are calculated in shallower waters, where the geothermal gradient is low and deformation is less prevalent;
- The amount of hydrate and free gas estimated over the studied area were $7.21 \times 10^{11}$ m$^3$ and $4.1 \times 10^{10}$ m$^3$, respectively;
- An inverse correlation between gas-phase concentrations and geothermal gradient is recognized. Low gas hydrate and free gas concentrations coincide with high values of geothermal gradients over the studied area;
- An extremely high geothermal gradient close to the trench was calculated, reaching values up to $190\ °\text{C·km}^{-1}$, caused by the subduction of the CR at the CTJ, altering the stable PT conditions for the gas hydrate, which promotes its dissociation and upward migration, and fluid escapes;
- High heat flow, tectonic faulting, and vigorous fluid advection may be important factors for hydrate and gas reservoir distribution offshore Taitao Peninsula;
- The CTJ is an important methane seepage area and should be the focus of novel geological, oceanographic, and ecological research.

**Author Contributions:** Conceptualization, L.V.-M. and I.V.-C.; formal analysis, L.V.-M. and I.V.-C.; funding acquisition, I.V.-C.; investigation, L.V.-M. and I.V.-C.; methodology, L.V.-M., I.V.-C., J.P.B., U.T., F.F., M.G., and S.C.; software, L.V.-M., I.V.-C., J.P.B., and U.T.; supervision, J.H.B.; visualization, L.V.-M. and J.P.B.; writing—original draft, L.V.-M.; writing—review & editing, I.V.-C., J.P.B., U.T., F.F., M.G., J.H.B., and S.C.

**Funding:** This research was funded by CONICYT- Fondecyt de Iniciación, 11140216.

**Acknowledgments:** Special thanks are due to Steven Cande and Stephen Lewis, who acquired the openly available data (http://www.ig.utexas.edu/) of R/V Robert Conrad Cruise RC2901. Lucía Villar-Muñoz acknowledges tenure of a DAAD scholarship for her postgraduate research and is grateful to the founders of GMT (Wessel and Smith). We are very grateful to Daniela Lazo and Rafael Santana, who contributed to the writing process.

**Conflicts of Interest:** The authors declare no conflict of interest. The funders had no role in the design of the study; in the collection, analyses, or interpretation of data; in the writing of the manuscript, or in the decision to publish the results.

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
