# Peer review of "Gas Hydrate Estimate in an Area of Deformation and High Heat Flow at the Chile Triple Junction"

_geosciences, doi:10.3390/geosciences9010028_

Reviewer 1 Report

This paper does a good job of pointing out the development of hydrate/free gas systems in an unusual structural and tectonic setting. The topic is interesting and the paper presents good results. There are issues discussed below that could be addressed to improve the presentation.

1] My primary issue is that the analysis is conducted in a setting that has many influential factors for hydrate and gas, but it is not clear from the paper how these can be used to assess the effects from this setting. As the paper points out, there are earthquakes, oversteepened slopes, high heat flow and subsurface fluid flow, all of which are important to hydrates. However, in order for this setting to be exploited, the specific the details of hydrate and gas concentrations need to be related to these features more specifically. In order to speculate on the potential for this setting to release the gas catastrophically, there needs to be more specific details about what these processes can do to hydrates and free gas in order to evaluate if there is a significant hazard. As it is, there are quite a few assumptions about subsurface fluid flow, etc., which may or may not have significant capacity to cause hydrate/gas concentration or rapid methane release, and may or may not be at work here.  In the paragraph beginning at line 273, there is a great deal of speculation regarding faulting and subsurface flow that has no firm basis. The few faults that are interpreted in Figure 3 do not appear to correlate very directly to hydrate and gas concentrations shown in Figure 4. Without knowing the source of the gas and more about how it is migrating in the subsurface, I would not argue that it is related to possible processes proximal to the spreading center as this paragraph tries to do. It looks like the highest hydrate/gas concentrations are located over an anticline, which is not too surprising. It is hard to see from these data how specific effects in this setting are affecting hydrate/gas concentrations. The results look similar to what could be expected across a profile far from the triple junction. It is also hard to understand hydrates/gas in this setting without some understanding of methane sources. See the third point below. There are obviously many unknowns that cannot be constrained here, especially regarding subsurface fluid flow, methane sources, etc.; however, there needs to be some more direct evidence and rationale for why this setting has affected the hydrate and gas system in specific ways, and what it reveals about hydrates.

2] Secondly, the main message is not clear. Is the message that despite relatively low concentrations of gas and hydrate, the tectonic setting has more potential for methane release (i.e. hazard) than other settings with more hydrate and gas? Is it that this high-heat flow and presumed high advective flow setting produces unusually large or surprisingly large amounts of hydrate/gas? Is it the large total volume of hydrate/gas despite the low concentrations, i.e. a surprisingly wide distribution in this setting? Is it the inverse relation between heat flow and hydrate/gas concentration? Is it a warning about possible greenhouse gas emissions to the atmosphere? The paper leaves me wondering what the overall significance of these results is. Is this setting just a one of a kind place that is interesting, but it is not particularly important for studies of hydrates broadly, or does it tell us something with broad application that is more apparent here than most other settings? The paper would be much stronger if there were fewer potential topics raised, but not fully developed, and a single (or possible two) message was clearly developed as the main focus.

3] Thirdly, there is a missed opportunity at this setting to consider abiotic methane, methane produced from serpentinization, which occurs in the upper mantle above young ocean crust. I don’t know much about it, but it probably should be considered. I don’t think drilling found any (or even looked for) evidence for abiotic methane here, but given that abiotic methane peaks in crust ~ 2 m.y. old, it may be, or have been, produced beneath the slope and an important consideration for hydrate distribution across the slope. This may not be significant, but should probably be considered and discussed.

4] The text needs to be edited to fix some incorrect or missing words. There were a number of places in the text with usage issues. I have marked some of them on the manuscript pages, but this should be examined throughout.

I have marked the manuscript with a number of issues that need to be addressed.  Here are a few of them that need to be considered.

Line 168: How is the theoretical Vp curve calculated? This is never defined, referenced or described. It is critical to the assessment of hydrate and gas concentrations and not straightforward. How are considerations such as lithology, porosity, etc. considered? Can uncertainty be estimated? Velocity estimates from data over rough seafloor are challenging. What are the uncertainties related to two-dimensional imaging with possible ocean currents over rough seafloor? This is probably not quantifiable, but some explanation of potential sources of error in velocity estimates and their likely impact on results should be included.

Line 268: In lines 208-211 the authors point out that hydrate concentrations are high above regions with low free gas, and vice versa. On line 268, the claim is that hydrate and gas are inversely correlated with heat flow. How can this be? They can’t all three be inversely correlated.

Line 296: It is hard to know what to make of the concentration estimates in this paragraph. I presume that what I’m supposed to learn from this is that the values are high enough and close enough to the sea surface that it could reach the sea surface if methane flux from the source was a bit higher. This is very hard to interpret without knowing methane flux at the seafloor related to these numbers, what sort of flux could be expected from methane release from an hydrate dissociation event suggested in this paper, as well as other oceanographic considerations, etc. This concept is intriguing, but not very well developed here.

Line 303: The list of reasons for low gas hydrate concentrations is not developed. This is mostly just a list without any discussion of how these factors lead to low concentrations and their potential impact in this setting.

Overall I like the paper and it will be a good contribution, but I think it could be improved to be more effective and have a bigger impact. I encourage the authors to consider these comments and revise the paper accordingly.

Author Response

Response to Reviewer 1 Comments

Point 1: Thank you very much to point us out about the speculation problems. Regarding the relationship between geological settings and gas hydrates occurrence, as suggested were not specified in this article, we would like to stress that the current paper is focused more on the analysis of the spatial distribution, concentration, estimate of gas-phases and geothermal gradients. From line 336 in the edited version of the manuscript (related to line 273 of your comments), we improved the text adding information about faults and fractures on the seismic profile.

Point 2: Many thanks for your highlight. Our work is mainly focused on to estimate gas hydrate quantities stored around the CTJ area, due to the special settings present here (high seismicity, high geothermal gradient, high fluid circulation, etc) promoting potentially dissociation and escape of methane to the seafloor. Eventually this knowledge will be essential in the geohazard assessment. In fact, our results deal with many topics as you pointed out, and they are summarized (the key points) in the abstract and conclusions. We already improved our conclusions considering your suggestions.

Point 3: We considered your point and we improved our discussion. This help us to understand a bit more the possible origin of the methane. Many thanks!

Point 4: According to your issues that need to be addressed (line 158, 268, 296, 303), we have modified and improve it (figures and text) in order to explain better our ideas.

Reviewer 2 Report

I have major concerns regarding this manuscript:

1) The velocity analysis is only described very briefly although it is the basis for the gas hydrate and gas estimates. Not a single velocity-depth profile is shown! The derived velocity-depth along the profile should be shown including estimates of uncertainty.

2) The calculation of temperature gradients is not explained in depth. It is unclear how they got the temperature at the seafloor and the temperature at the BSR. Also it would be much better to calculate heat flow instead of just temperature gradients.

3) In Villar-Munoz (2014) heat flow along the seismic profile 751 has already been calculated. In Villar-Munoz et al. (2014) the BSR is not continouos along Profile 751 (see Fig. 7). Why is it now? There is no in depth discussion with results obtained in the previous paper.

4) Why is there no in-depth discussion with results obtained in ODP Leg 141? Why is not this transect of boreholes along seismic profile 745 used to ground-truth the method they present in this paper for seismic profile 751?

5) No uncertainties of any of the estimates are shown. This is not good scientific practise.

6) One minor point: why do they really need 75 (!) references?

Author Response

Response to Reviewer 2 Comments

1)      We included velocity-depth profiles in Fig. 3 and improved the text according your suggestions.

2)      We improved the Methods-d) section and we explain better our calculations as you suggested. We would clarify that both, heat flow and geothermal gradient, are important in our work, because the geothermal gradient is used in calculations related to gas hydrates as well as we used this gradient to confirm heat flow values already reported in literature. In fact, the heat flow and geothermal values that we obtained are consistent with the geological setting of the study area.

Since a regional heat flow overview has already done by Villar-Muñoz et al. (2014), we focus now mainly on the local heat flow and geothermal gradient.

3)      In the present work, we performed a detailed seismic analysis in order to add new information (e.g. velocity modelling and geothermal gradient) to the Villar-Muñoz et al. (2014) article. Due to this processing, we could identify much better the BSR on the seismic profile (after a detailed velocity modelling).

4)      We added discussions about Leg 141, as you required.

5)      We added sensitivity values in the method section.

6)      As we intend to reach a wide scientific spectrum (e.g. geology, biology, oceanography, ecology) and the scientific interest in this special place, we add a large number of research is reported in literature that are significant and relevant for the present study.

Reviewer 3 Report

See file attached

Author Response

Response to Reviewer 3 Comments

Dear Reviewer 3:

We would like to acknowledge your comments, advices and corrections for our manuscript.

We considered all your suggestions and we modified/improved the manuscript and figures, as you asked for, as well as the methodology section in which you wanted more information and detail. Besides, we edited the sentences, as you required (lines: 141, 143, 144, 145, 148, 154, 182, 189, 193, 194, 197, 205, 213, 218, 234 and 250).

Many thanks for your time and advices!

Villar-Muñoz, et al.

 Round  2

Reviewer 1 Report

The authors have considered my previous concerns and have made improvements. The current version is certainly better. There are a few things that I would like to see before publication, which would make this easier for the readers and a better overall manuscript.

1) There are many issues with missing words, wrong tense, or plural vs. singular. These need to be fixed to make it more readable.

2) I'm still puzzled by the statements on lines 209 and 219-222 of the new version. These seem contradictory and confusing. I have read it many times and don't see what I may be getting wrong. I pointed this out in my past review and it does not seem to have changed. This is very confusing and the point becomes lost unless it is clear which is meant.

3) Line 233: The observation about heat flow is written as if it is another new finding from the study. Because heat flow is linearly related to thermal gradient, this is essentially saying the same thing twice. There is nothing wrong with discussing heat flow, but this should be written to reflect that there is really no new information.

4) Lines 277-281: The statement here is written as if this is an important finding of the study, i.e. "associated to". However, this is the basis of the analysis. The deviations from the reference define either gas hydrate or free gas concentrations. First, what this does not consider is other possibilities for velocity changes, such as lithology, compaction, fractures, etc. Secondly, as written there is an implication that "high" gas concentrations can be detected when the reality is that velocity is only sensitive to free gas at low concentrations (1-2%).

5) Lines 288-304: This section first says these factors "could" (289 & 298) or "can" (294) control hydrates and free gas. The first statements are fair because it is possible and certainly will have some role, but not well proven here. But then what follows is the conclusion that these same factors "are the main factors" (304). This is not based on anything more than what was stated above and is not proven here.There is nothing presented (or much known, probably) about gas sources, production and migration to know that the observed distribution of hydrate and free gas is a function of the processes that they are attributed to in the statements in this paragraph. As I mentioned in my previous review, abiotic methane is probably a source. This is now discussed briefly, but not in terms of how it may affect methane production and distribution across the slope. Abiotic production is not uniformly distributed with distance from the ridge.

6) Lines 288, 294, and 354: There is an implicit assumption made in this analysis that needs to be acknowledged. The assumption is that prior to the effects that are causing hydrate dissociation, there was an equal amount of hydrate to begin with and that the dissociated hydrate was lost. What is actually more likely is that hydrate concentrations had variations due to production, and hydrates that dissociated released free gas that migrated up into the hydrate stability zone and reformed as hydrate. Without a good model of the processes described here, the statements here have pretty limited value.

Author Response

Response to Reviewer 1

 Dear Reviewer :

We would like to acknowledge your comments, advices and corrections for our manuscript.

We considered all your suggestions as following:

1.      We modified/improved the English grammar of the manuscript as you asked for.

2.      We improve the statements on lines 209 and 219-222 of the last version, as you required.

3.      We changed Line 233 of the last version and deleted the information about heat flow because was not part of our results as you suggested.

4.      We are agree with your comment that the change of velocity can be attributed not only to gas hydrate and to free gas presence, but also it can be related to other geological conditions. Anyway, as described in the paper, the presence of two reflectors that can be interpreted as BSR and BGR supports our interpretation. Moreover, the low velocity (1290 m/s) below the BSR can be only associated to the presence of free gas, which is better explained now in the manuscript.

5.      We have attempted to make the statement in the line 304 of the last manuscript according with the interpretations further up in the text.

6.      Regarding lines 288, 294, and 354 of the last manuscript, we include new sentences in the Discussion and Conclusion section to improve them. We now explain better the gas-phases (gas hydrate and free gas) distribution taking in consideration your suggestion.

Many thanks for your time and advices,

Villar-Muñoz, et al.

Reviewer 3 Report

I am happy with the revised version and the efforts by the authors.

Author Response

Dear Reviewer,

Thank you very much for your advices. We improve the english grammar of the manuscript as you suggested.

Sincerely,

Villar-Muñoz et al.